# High β-Glucans Oats for Healthy Wheat Breads: Physicochemical Properties of Dough and Breads

**DOI:** 10.3390/foods12010170

**Published:** 2022-12-29

**Authors:** Valentina Astiz, Luciano Martín Guardianelli, María Victoria Salinas, Carla Brites, María Cecilia Puppo

**Affiliations:** 1EEA Cesareo Naredo, Instituto Nacional de Tecnología Agropecuaria (INTA), Buenos Aires 6417, Argentina; 2Centro de Investigación y Desarrollo en Criotecnología de Alimentos—CIDCA, Facultad de Ciencias Exactas-UNLP-CONICET, La Plata 1900, Argentina; 3Instituto Nacional de Investigação Agrária e Veterinária, I.P. Av. da República, Quinta do Marquês, 2780-157 Oeiras, Portugal

**Keywords:** wheat, oat, β-glucans, dough rheology, bread quality

## Abstract

Bread is a highly consumed food whose nutritional value can be improved by adding an oat flour (*Avena sativa* L.-variety Bonaerense INTA Calen-Argentina) to a high-industrial quality wheat flour (*Triticum aestivum* L.). This cultivar of oat contains high amounts of β-glucans, which act as a prebiotic fiber. Wheat flour was complemented with different amounts of oat flour (5, 15, and 25%). A contribution of hydrophilic components from oat flour was evident in the oat–wheat mixtures. At the same time, the high content of total dietary fiber led to changes in the rheological properties of the dough. Mixtures with a higher proportion of oats showed an increase in alveographic tenacity (stiffer dough), higher stability, and a lower softening degree in farinographic assays. The dough showed significant increases in hardness and gumminess, without significant changes in cohesiveness, i.e., no disruption to the gluten network was observed. Relaxation tests showed that the blends with a higher oat content yielded 10 times higher stress values compared to wheat dough. Analysis of the oat–wheat breads showed improvements in nutritional parameters, with slight decreases in the volume and crust color. The crumb showed significant increases in firmness and chewing strength as the amount of oats added increased. Nutritional parameters showed that lipids, dietary fiber, and β-glucans were significantly increased by the addition of oats. Sensory analysis achieved high response rates with good-to-very good ratings on the hedonic scale set. Thus, the addition of oats did not generate rejection by the consumer and could be accepted by them. Breads with wheat and oats showed nutritional improvements with respect to wheat bread, since they have higher dietary fiber content, especially in β-glucans, so they could be considered functional breads.

## 1. Introduction

Oat (*Avena sativa* L.) is a cereal crop grown worldwide for human and animal feed. In addition, compared to other crops, oats are more suitable for production in marginal environments, including cold, wet climates and soils with low fertility [1]. The main areas of oat cultivation are between the latitudes of 40° and 60°, i.e., northern America, Europe, and Asia; a small proportion of the world oat production is located in the southern hemisphere, i.e., South America, Australia, and New Zealand [2]. Unlike wheat (*Triticum aestivum* L.), oats do not have the ability to develop gluten; however, they possess β-glucans, which are considered a prebiotic soluble fiber with beneficial health effects. The β-glucans are non-digestible polysaccharides of natural origin found in different organic sources, such as corn grains, yeasts, bacteria, and algae, among others. They are important components of fibers and contain unbranched polysaccharides consisting of β-D-glucose units linked through glycosidic bonds β (1 → 4) and β (1 → 3) in cereals and glycosidic bonds β (1 → 6) in fungal sources. Glucans are generally concentrated in the bran, the aleurone, and sub-aleurone layers [3]. The highest content of β-glucans is found in barley and oat grains, compared to other cereals/pseudocereals such as wheat, rice, buckwheat, millet, and amaranth [4]. Nutritional improvements associated with the benefits of incorporating β-glucans in the diet have been evidenced, including cholesterol reduction, glycemic control, positive effects on the cardiovascular system, and antitumor and immunomodulatory effects [4,5,6]. Although β-glucans are important molecules from the nutritional point of view, techno-functional aspects such as the rheological properties of dough and bread quality are also relevant. Londono et al., (2015) [7] have studied the role of these carbohydrates in the technological quality of wheat-free and wheat dough. These authors found that the elastic properties of oat dough added with wheat gluten were negatively affected with the increase in β-glucans content. They attributed this unfavorable effect to the concentration and viscosity conferred by β-glucans, since using lower molecular weight and less viscous β-glucans in a lower concentration counteracted the negative impact.

β-glucans have also been cited as the active compounds of soluble dietary fiber, responsible for playing a fundamental role in intestinal physiology due to prebiotic effects [8]. Therefore, the aim of this work was to evaluate the effect of the addition of an oat flour with high β-glucan content on the technological quality of wheat flour-based dough and breads. 

## 2. Materials and Methods

### 2.1. Materials

Wheat flour (*Triticum aestivum* L.) was obtained by milling Buck Meteoro wheat grains provided by Chacra Experimental Integrada INTA-Barrow (INTA, Tres Arroyos, Provincia de Buenos Aires, Argentina). This wheat flour presented 13.7 ± 0.1% moisture, 14.0 ± 0.1% proteins, 1.3 ± 0.1% lipids, 2.82 ± 0.06% total dietary fiber, 0.55 ± 0.01% ash, and 67.6% carbohydrates different from fiber. 

Oat grains (*Avena sativa* L.) of the variety Bonaerense INTA Calen were provided by Chacra Experimental Integrada INTA-Barrow (INTA, Tres Arroyos, Argentina). These seeds were milled using a Perten 120 mill (Perten, Hargersten, Sweden) with a mesh size of 0.5 mm, resulting in an oat flour that complies with the specifications of the Argentinean Alimentarius Codex, Chapter IX, Article 698 [9]. The oat flour presented 11.4 ± 0.1% moisture, 16.2 ± 0.1% proteins, 3.5 ± 0.3% lipids, 1.8 ± 0.1% ash, 4.0 ± 0.2% β-glucans, 10.98 ± 0.45% total dietary fiber, and other carbohydrates (53.7%). This experimental oat flour has a higher β-glucans content than other commercial brands. 

Other ingredients used were sodium chloride (Celusal, Industrias Químicas y Mineras Timbó S.A., Buenos Aires, Argentina), fresh yeast (CALSA, Tres Arroyos, Argentina), and distilled water. Blends with wheat flour complemented with oat flour (5, 15, and 25% wheat flour basis) were prepared. A control (C) sample of wheat flour was included.

### 2.2. Methods

#### 2.2.1. Alveographic Characterization

Alveographic tests were performed in a Chopin alveograph (Villeneuve-la-Garenne Cedex, 92390, France) according to AACC [10]. The parameters obtained for the different doughs were: tenacity (P), extensibility (L), P/L ratio, and deformation energy of the dough (W). Tests were performed in duplicate.

#### 2.2.2. Farinograph Assays

Farinographic assays were conducted in a 300 g Brebander farinograph (Duisburg, Germany). The parameters obtained were: water absorption (Wabs), development time (DT), stability (ST), and softening degree (SD), according to AACC [10].

#### 2.2.3. Dough Preparation

Wheat flour (control sample: C) and blends of wheat flour with different proportions of oat flour (5, 15, 25% *w*/*w*) were mixed in the absence of yeast with 1.5% salt and distilled water in the quantity according to farinographic water absorption (Table 1). The farinographic development time was used as the kneading time for dough preparation. 

The solid ingredients (flours and salt) were first placed in a Kenwood Major planetary mixer (Kenwood Major, Milano, Italy) and mixed in a dry form. After that, distilled water was added and kneaded for the first minute at 50 rpm; then, the ingredients were mixed at 90 rpm till achieving the kneading time. The dough formed was removed from the mixer and allowed to rest for 10 min at 20 °C covered with plastic film and manually laminated.

##### Moisture and Water Activity of Oat–Wheat Dough

The moisture of the dough was determined at 105 °C [10], and water activity (aw) was measured by the dew-point method at 25 °C on Meter AquaLab series 4 TEV equipment (Decagon Devices Inc., Washington, DC, USA). The assays were performed in triplicate.

##### Rheological Properties of Oat–Wheat Dough

Dough texture parameters and tension–relaxation tests were performed using a TA.XT2i texture analyzer (Stable Micro Systems, Surrey, UK) with a 25 kg total load cell. A 75 mm diameter cylindrical probe was utilized. Cylindrical disks of dough (height = 1 cm, diameter = 3 cm) were prepared for both assays, and tests were performed in duplicate.

##### Texture Profile Analysis of Oat–Wheat Dough

Each disk of dough was subjected to a double compression cycle using a deformation of 40% of the original height. The test speed was 0.5 mm/s, and 5 s passed between the two cycles. Twenty discs were tested for each formulation. Hardness, cohesiveness, gumminess, consistency, springiness, and adhesiveness were calculated from the curves [11]. The assays were performed in duplicate.

##### Relaxation of Oat–Wheat Dough

The dough disk was compressed to 40% of its original height at a crosshead velocity of 0.5 mm/s for 1200 s. Dehydration of the dough was avoided by placing semi-solid silicone on the edges. The stress–relaxation curves were fitted with OriginPro 8 software (OriginLab Corporation, Northampton, MA, USA) using a second-order exponential decay regression. Tests were performed in triplicate for each formulation. A generalized Maxwell model [12] of 2 Maxwell elements with a parallel residual spring was applied [13] (Equation (1)).
(𝑡) = 𝜎_1_𝑒^−𝑡/𝑇1^ + 𝜎_2_𝑒^−𝑡/𝑇2^ + 𝜎_𝐸_(1)
where:σ_i_ = pre-exponential factors (equivalent to a σ_1_ y σ_2_);T_i_ = relaxation times, defined as the ratio between the viscous and elastic component (η_i_/E_i_);σ_E_ = represents the equilibrium effort.

#### 2.2.4. Bakery Properties of Oat–Wheat Dough

##### Dough Preparation

Dough was prepared according to the “Dough preparation” section, but 3% fresh yeast dissolved in part water was added.

##### Dough Fermentation

Once the dough was prepared, spherical portions were cut (50 g) and placed in graduated cylinders (diameter: 5 cm, volume: 500 mL) equipped with a mobile plunger. Cylinders were introduced in a fermentation chamber at 30 °C. The increase in volume (ΔV) was measured as a function of time for 4 h. The ΔV vs. time plots were fitted with the Sigmaplot 10.0 program using Chapman’s 3-parameter model (Equation (2)).
ΔV = a × [1 − exp (−b × t)] c(2)
where:ΔV = dough volume increment (mL);t = test time (min);a = Vmax = maximum volume occupied by the dough (mL);b = rate constant for the increase in dough volume (min^−1^);tf= fermentation time, the time needed for reaching ¾ Vmax (min);c = parameter related to the inflection point and the shape of the curve.

##### Breadmaking Process

The dough was rolled, allowed to rest for 15 min, cut into 90 g buns, and shaped in a bread machine (MPZ Zambon, Buenos Aires, Argentina). The pieces were fermented at 30 °C according to the optimum fermentation time and then baked in a convection oven (Ariston, Buenos Aires, Argentina) (20 min, 200 °C). Bread quality was evaluated 90 min after baking.

#### 2.2.5. Baking Quality

##### Specific Volume

The volume of 5 loaves of each formulation was determined by displacement of rapeseed in a pan-volumeter. The initial volume occupied by the seeds was measured, and then a loaf of known weight was added, and the new volume achieved was read. The seed displacement produced was directly proportional to the volume of the bread. The specific volume was calculated as the ratio of the volume to the weight of the loaf.

##### Color of Crust

The color of the bread crust was measured with a colorimeter (Chroma Meter CR 400, Konica Minolta, Osaka, Japan). The brightness (L*), a* (+, red; −, green), and b* (+, yellow; −, blue) color coordinates were determined according to the CIELab space system. Eight loaves of each bread formulation were used for color measurements, and five fields per loaf were assayed. The browning index (BI) was calculated according to equations Equations (3) and (4):x = (a* + 1.75 L*) 5.645 L* + a* − 3.012 b*(3)
BI = 100 (x − 0.31)/0.172(4)
where:a*: position between green and red;b*: position between blue and yellow;L*: brightness.

##### Crumb Texture

Crumb texture was determined as described for the dough. The following parameters were determined on the crumb: firmness, cohesiveness, chewiness, and springiness.

#### 2.2.6. Bread Composition 

Composition was determined for all breads. Moisture content was assayed at 105 °C by the indirect method, proteins by the Kjeldahl method (f = 5.71), lipids by the Soxhlet method, and ashes by the direct method using a muffle at 550°C. The β-glucans content was determined using the Megazyme (1–3) (1–4) β-D-glucan assay kit, and the total dietary fiber was calculated by the enzymatic-gravimetric method according to AACC [10].

#### 2.2.7. Sensory Quality of Breads

A sensory analysis of the breads was designed with a non-trained panel of at least 45 people. A 9-point hedonic scale was utilized. The attributes of appearance, texture, flavor, and overall acceptability were evaluated.

#### 2.2.8. Statistical Analysis

Results were calculated by simple ANOVA using the software InfoStat [14]. Fisher’s LSD test at a significance level of 5% was utilized for comparing means.

## 3. Results and Discussion

### 3.1. Dough Alveographic and Farinographic Assays

Table 2 shows the values obtained for the alveographic and farinographic parameters of the wheat flour–oat flour blends. As the percentage of oat flour increased, there was a significant increase in dough tenacity (P) and a decrease in extensibility (L), with a consequent increase in the P/L ratio. All blends showed a P/L ratio >1 associated with more tenacious dough, which could affect the expansion of the dough during leavening and the future volume of the bread. In contrast, the alveographic work (W), related to baking strength, did not show significant differences with the addition of oat flour. It can be concluded that the strength of the dough was not affected, unlike the tenacity, which could result in a harder dough. Popa et al. [15] found that the addition of oats caused some technological problems; the authors observed a decrease in dough extensibility due to the effect of the high content of total dietary fiber, as it was detected in this work. This behavior is probably due to the high viscosity of the β-glucans [7].

On the other hand, it was observed that dough quality was significantly improved by the addition of oat flour (Table 2). The water absorption (WA) to reach the optimum consistency (500 BU) of the dough was significantly higher as the percentage of oats added increased. Similar results were reported by different authors [16,17,18,19] when they used other oat varieties with a lower content of β-glucans. This behavior could be attributed to the higher protein content of the mixtures with respect to the control wheat flour (Buck Meteoro); due to INTA, Calen oat has a high content of these macromolecules (18.3 ± 0.1%). It could also be due to the high content of total dietary fiber of this oat (10.98 ± 0.45%) in comparison to the value of Buck Meteoro wheat flour (2.82 ± 0.06%) [15,18,20].

The development time (DT) associated with the time in which the dough reached the ideal consistency did not change with the addition of oat flour (Table 2), but in all cases, it was high (>24 min). The time in which the dough is maintained at the ideal consistency is the stability (ST). This property also showed no statistical differences among the formulations; in all formulations, the stabilities were excellent for baking (>28 min). Finally, the softening degree (SD), which measures how many BU the curve falls from the ideal consistency, decreased significantly as the amount of oat flour increased in the blend. In general, for wheat flours, as farinographic stability increases, the softening degree decreases. In this case, the opposite behavior was observed: the dough stability decreased in a non-significant way, i.e., it remained practically constant, and the softening degree steadily decreased. This behavior, which is positive considering the baking aptitude, could be attributed to the presence of some components (proteins and β-glucans, among others) in the oat. The results indicate that the addition of oat positively affected dough rheology. According to Zhang et al. [16], the high content of β-glucans could be responsible for the increase in water absorption and the improvement observed during kneading. Different authors who have studied oat addition at percentages between 5 and 25% also reported an increase in water absorption during kneading. This capacity was mainly determined by the presence, in the fiber structure, of a large number of hydroxyl groups that interact with water through hydrogen bonds [15,17,18,21,22]. In contrast to our results, Mis et al. [22] evaluated breads with an oat addition of between 5 and 25% and obtained higher farinographic stability values, which was attributed to the kind of structure of the fiber present in oats.

The improvements observed in the alveograms and farinograms for the mixtures studied were also found by other authors [18,21,23]. They attributed this behavior to the high fiber content of oats, especially soluble fiber, which would positively modify the development of the gluten network, giving it greater consistency. Djordjević et al. [24] reported that enriching flours with dietary fiber from oat resulted in an increase in dough consistency and viscosity, due to the crosslinking of optimally hydrated cellulose and hemicellulose chains that originated from the predominantly insoluble fibers added.

### 3.2. Evaluation of Hydration Properties of Dough

Table 3 shows the water activity (aw) values of the different doughs. The control dough had a value of aw = 0.978, and for the rest of the formulations, it was also high and statistically similar (>0.979), indicating that these doughs are systems with high water availability, and that the addition of oat flour did not modify this property.

In the case of moisture, a value of 55.2% could be observed for the wheat dough, which is significantly higher than that of the rest of the formulations. In addition, the moisture content decreased with the increasing amount of oat flour. This trend was inverse to that observed with farinographic absorption (Table 2), which is attributed to the high fiber content present in oat flour. Dough with oat would form a matrix that requires a greater amount of water that binds to wheat, oat proteins, and to the different hydrophilic components, such as β-glucans, pentosans, soluble fiber, and other soluble carbohydrates. Although higher water absorption was obtained, the moisture content of the wheat–oat dough was much lower than that of the wheat dough (C). This behavior could be due to the fact that the excess of water would be part of the bound water; it is not water in a free state and therefore could not be determined by heating. Therefore, this water would be structurally bound to the different hydrophilic components mentioned above; that is, it would be constitutional water that may contribute to structuring the dough matrix. Wang et al. [25] evaluated the effects of the incorporation of oat β-glucan (between 1% and 5%) on the hydration properties of wheat flour dough. They observed, as in our case, an increase in the farinographic water absorption of the dough supplemented with high levels of a commercial soluble oat β-glucan. For these authors, significantly less bound water and more immobilized water than the control dough were observed, suggesting a water deficit for gluten network development, whereas water molecules attached to dietary fiber tend to be in an immobilized state. 

### 3.3. Texture Profile Analysis of Dough

The texture profile analysis of the different doughs is shown in Table 4. Doughs were harder as the oat content increased, resulting in a value four times higher for the dough with 25% oat flour (10.9 N) with respect to the control wheat dough (2.7 N). Cohesiveness did not show a definite trend, and gumminess showed the same behavior as hardness, going from values of 2.1 N to 7.3 N in the dough with higher oat content (25%). Hager and Wolter [26] found that a dough with 100% wheat flour presented higher hardness than one with 100% oat flour. The type of interaction established between the components of the two flours (proteins, starch, and fiber) favors the formation of more structured dough compared to single flours.

Consistency showed a significant increase as the percentage of oat increased, following the same trend as hardness. Adhesiveness decreased significantly with the addition of oat flour (Table 4). Other authors have reported increases in the adhesiveness of dough by the incorporation of other flours rich in fiber and proteins, such as soybean flour [27] and chickpea flour [28], which is opposite to that which was found in this work. This is probably because the chemical nature of the oat flour components increases the water absorption of the flour, forming a harder and less adhesive dough. These compounds are highly hydrophilic molecules (β-glucans and others) capable of forming strong hydrogen bonds with water. This means that water is adsorbed to these components and incorporated into the gluten matrix, with no ability to migrate to the dough surface.

### 3.4. Relaxation of Dough

The relaxation parameters of the dough are shown in Table 5. The relaxation time (T) behaves inversely to the elastic modulus (E) and is related to the degree of relaxation; higher values of T are associated with a more viscous behavior with respect to the elastic one, and the dough will relax more. Moduli E1 and T1 govern the relaxation at the beginning of the deformation and are attributed to the orientation of the small-sized molecules, and E2 and T2 (intermediate zone) represent the relaxation of the polymeric molecules. Since dough relaxes in greater proportion at the beginning, T1 is generally greater than T2. If the stress does not change with deformation, and equilibrium is reached, the E3 term predominates, which represents the stored energy.

All doughs presented higher-order values of E2 compared to the E1 and E3 moduli. This suggests that the gluten polymeric proteins that relax in zone 2 (represented by E2) are contributing greatly to the elasticity of the dough. 

Dough with 5% oats does not modify the relaxation parameters; however, the addition of 15% and 25% oats results in a predominance of the elastic component without a clear trend in the relaxation times (T1 and T2). In dough with 25% oats, all moduli (E1, E2, and E3) were significantly higher than for the rest of the samples, suggesting the formation of more elastic dough. These results are in agreement with those cited by Bigne et al. [29], who studied wheat flour blends with up to 20% chickpea flour.

### 3.5. Fermentation of Dough

Fermentation curves (volume increase as a function of time) obtained experimentally for all dough are shown in Figure 1. It can be observed that the addition of oat flour originated curves with different ∆V with respect to the control (C). Dough C had a greater increase in volume with respect to the oat-containing dough, the latter reaching a plateau at shorter fermentation times. A significant decrease in the volume of fermented dough was observed with respect to the wheat dough, and it was very pronounced for the sample with 25% oats.

The following values of fermentation times, calculated as ¾ Vmax, were obtained: 44.5 min (C), 36.9 min (5%), 33.5 min (15%), 17.0 min (25%). A similar behavior was found by de Erive et al. [30] when adding oat soluble fiber (10, 12 and 14%) of high β-glucan content (70%) in wheat doughs. They found that doughs with high levels of fiber did not expand sufficiently during fermentation, indicating insufficient carbon dioxide production due to the lack of weakly bound water to maintain normal yeast metabolic activity and/or the poor gas-holding capacity of the bread dough.

The parameters obtained from Equation (2) are shown in Table 6. Parameter a (maximum volume) decreased with the addition of oat flour regardless of the percentage of addition. Parameter b, which is related to fermentation rate, showed an increase with the oat flour content and was significant for the 25% level. Therefore, doughs with 25% oats provide components that accelerate fermentation, but the dough expands less than C. This tendency is associated with higher hardness and higher elasticity (TPA and relaxion assays) and could affect the volume of the baked goods. Finally, parameter c, which is related to the inflexion point and the shape of the curve, showed no significant differences between samples, suggesting that the way the dough ferments over time is dominated by the formation of the wheat gluten network. The variations observed could be attributed to the contribution of fiber by the oat flour that would interfere with the development of the gluten matrix, affecting its viscoelastic properties that are necessary for dough expansion, carbon dioxide retention, and alveoli formation [31].

### 3.6. Evaluation of Bakery Quality 

The results of specific volume (Vsp) for the obtained breads are shown in Figure 2A. The addition of oat flour generated a significant decrease in bread volume. The control bread presented a Vsp value of 3.8 ± 0.4 cm^3^/g. The formulation with 5% oats showed a decrease of 18% in Vsp, whereas in blends with a high oat content, this parameter decreased around 40%. In this sense, Litwinek et al. [19] evaluated 50:50 mixtures of oat:wheat flours and found a significant decrease in the Vsp of the breads. Since the weights of the pieces were not significantly different, the authors attributed this decrease to the absolute volume values of the breads. They assigned this behavior to the higher water retention during dough formation. Other authors have also observed a decrease in bread volume with the addition of up to 30% oats to replace wheat flour, attributing this to the different composition and grain size of the flours [32]. In another system, Salinas et al. [33] evaluated the effect of incorporating two European carob flours, pulp and germ, to wheat flour in an amount of up to 30%. These authors found a greater reduction in the specific volume of the breads when the pulp flour was incorporated (with a total dietary fiber content of 45%) compared to breads with the incorporation of germ flour (with 55.7% protein) of this legume; these results allowed the authors to infer that the detriment in the quality of the product is greater with the incorporation of certain components, such as fiber, versus others.

It should be clarified that all the aforementioned authors agreed that the volume losses were attributed to the dilution effect on gluten proteins and/or to the disruption of the gluten network caused by the new components.

Rieder et al. [21] also studied breads with added oat flour in proportions between 5 and 25% and found significant decreases in bread volume and, consequently, increases in crumb firmness as well. The difference in our work lies in the fact that we used an oat variety (INTA, Argentina) with a high β-glucan content (4.0%) whose baking behavior should be analyzed in depth.

Compared to wheat flour, oat contains a higher value (twice as much) of total dietary fiber, with 65% corresponding to insoluble fiber [20]. Djordjević et al. [24] mentioned that insoluble dietary fiber shows a negative influence on bread volume due to its hydration capacity. In our case, the total dietary fiber content of INTA-Calen oat was 10.98 ± 0.45%, four times more than that of wheat (2.82 ± 0.06%).

The Browning Index (BI) is a parameter that has shown a linear correlation with brown pigment content and has been useful for the evaluation of color changes in foods undergoing a Maillard reaction. In particular, in baking, it has been possible to observe variations in crust color during cooking due to modifications in the formulations used [34,35]. Figure 2B shows the Browning Index (BI) values of the different formulations of breads. Since the temperature and baking time were the same for all the breads, we can infer that the change in coloration is due to the ingredients of each formulation, i.e., the amount of oat flour added. Between the control and 5% breads, no significant differences were observed for the BI value, and the highest values were associated with the breads with a more brownish crust. As the oat content increased, the BI decreased by 22% and 30% for the 15% and 25% breads, respectively, suggesting that the crust became less browned. These results agree with those found by Litwinek et al. [19], who studied breads with 50% oat flour and 50% wheat flour and found a decrease in crust color relative to wheat bread. In this context, Tamba-Berehoiu et al. [36] found, when mixing wheat flour with up to 50% oat flour, that the color of the bread crust was lighter as the amount of oat flour increased, a phenomenon that could be related to the reduction in the Maillard reaction rate and the formation of dextrins, due to the reduction in the amount of water available for these reactions. On the other hand, Londono states that β-glucans increase viscosity, which probably slows down the diffusion of molecules having free amino and carboxyl groups with the consequent decrease in Maillard reactions [7].

### 3.7. Crumb Texture

Figure 3 shows the texture parameters (firmness, cohesiveness, chewiness, and springiness) obtained from the TPA of fresh crumbs of the control breads (100% wheat flour, C) and the formulations with 5, 15, and 25% oat flour. A significant increase in firmness (Figure 3A) and chewiness (Figure 3C) was observed as oat levels increased, whereas cohesiveness (Figure 3B) and springiness (Figure 3D) showed no significant differences between formulations, indicating that crumb integrity was not altered by the presence of oat components. 

Different authors found an increase in the firmness of wheat bread crumbs with increasing oat content (up to 50%), attributing this trend to a lower specific volume of the breads [19,21]. In contrast to our results, Djordjević et al. [24] found for oat breads that cohesiveness and elasticity decreased with respect to wheat bread.

### 3.8. Nutritional Value of Breads

The percentage composition values are shown in Table 7. It can be observed that the crumbs’ moisture was between 32.3–35.5%. The protein content was significantly similar (≈ 9.4%) with up to 15% oats added. Meanwhile, with 25% oats, a 0.56% decrease in the protein percentage with respect to the control was observed. Breads with the highest levels of oats (15 and 25%) presented a higher percentage of lipids. Ash levels did not show a clear trend with the variation in the oat levels in the bread. Total dietary fiber values showed a significant progressive increase as the percentage of oat flour in the formulation increased. This behavior is in line with the composition of the pure flours, showing that the lipid and TDF contents of the oat flour were higher than that of wheat, which coincides with data from the literature [19].

In the case of β-glucans content, a progressive increase was also observed with the addition of oat flour, reaching, at the maximum level (25%), an increase of 3.5 times the value for wheat bread. Litwinek et al. [19] achieved increases of 0.80 to 0.85% in breads with 50% oat flour. In several works, it was found that the addition of oat flour in baked goods, even in small amounts, produces modifications in the chemical composition of the final product [17,18]. On the other hand, other authors have found that the addition of oat caused an increase in the content of lipids, ash, β-glucans, and total dietary fiber compared to a bread made with 100% wheat flour [19,32]. Manthey et al. [37] found that soluble fiber accounts for 42% of total dietary fiber (TDF). This soluble fiber contains mainly β-glucans (64% glucose), neutral sugars (15% arabinose, 10% galactose, 1.5% mannose, 1.5% ribose, and 7% xylose), and uronic acids (0.1%). On the other hand, 58% of TDF is insoluble fiber and is composed of neutral sugars (20% arabinose, 2.5% galactose, 40% glucose, 2.7% mannose, 2.3% ribose, and 30% xylose), 0.5% uronic acids, and approximately 2.7% lignin. Breads with oats could be considered high in fiber according to the legislation in Argentina regardless of the percentage of addition, considering that 100 g of bread contains more than 6 g of dietary fiber [9].

### 3.9. Sensory Quality of Breads

Figure 4 shows the response percentages for each sensory attribute analyzed for the three formulations containing oat flour. For all formulations and attributes evaluated, the highest percentage of responses was found between points 7 and 8 of the hedonic scale. In the case of the attribute “appearance”, it was detected in 75% of 7–8 responses for the bread with 5% oats, 76% for the bread with 15% oats, and 66% for the formulation with 25% of oat flour, respectively. In the case of “texture”, the percentage of responses between 7 and 8 were 53% for the 5% bread, 44% for the 15% bread, and 40% for the formulation with 25% oat flour. “Flavor” showed 56% of responses between the mentioned values (7–8) for the 5% formulation and 53% for both the sample with 15% and 25% oat flour.

“Overall acceptability” also received the highest response percentages between items 7 and 8 of the scale and was, in all formulations, greater than 50%.

## 4. Conclusions

The mixture of an oat flour high in β-glucans, variety Bonaerense INTA Calen, with a wheat flour of good baking aptitude allowed for the obtainment of dough and breads of good technological and nutritional quality that were acceptable to consumers. A higher level of oats (25%) made it possible to obtain stable, less-softening, and harder dough compared to wheat. The mixture was also the hardest and most elastic, which resulted in a smaller increase in dough volume during fermentation together with a lower specific volume of the bread (25%). However, the bread crumbs with the highest amount of oats were equally cohesive and elastic but harder. Despite this, this mixture had the highest dietary fiber content, especially β-glucans, which are considered to be a fiber with a prebiotic effect. Regarding sensory analysis, high response percentages were observed with good-to-very good ratings for the selected hedonic scale, indicating that the addition of oats did not generate consumer rejection of the breads. Therefore, an addition of 25% oats to wheat flour would be a good alternative for making pre-mixes for obtaining breads of good technological quality and, potentially, prebiotic activity, due to their high content of dietary fiber.

## Figures and Tables

**Figure 1 foods-12-00170-f001:**
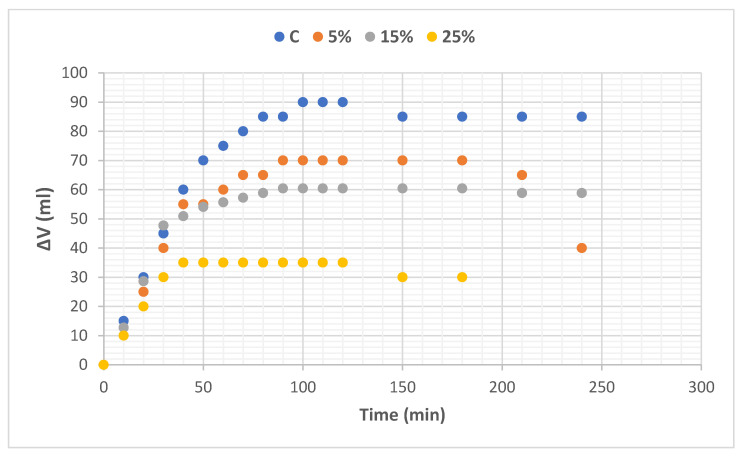
Fermentation curves of dough: wheat flour (control: C) and blends of wheat flour and oat flour at levels of 5, 15, and 25%.

**Figure 2 foods-12-00170-f002:**
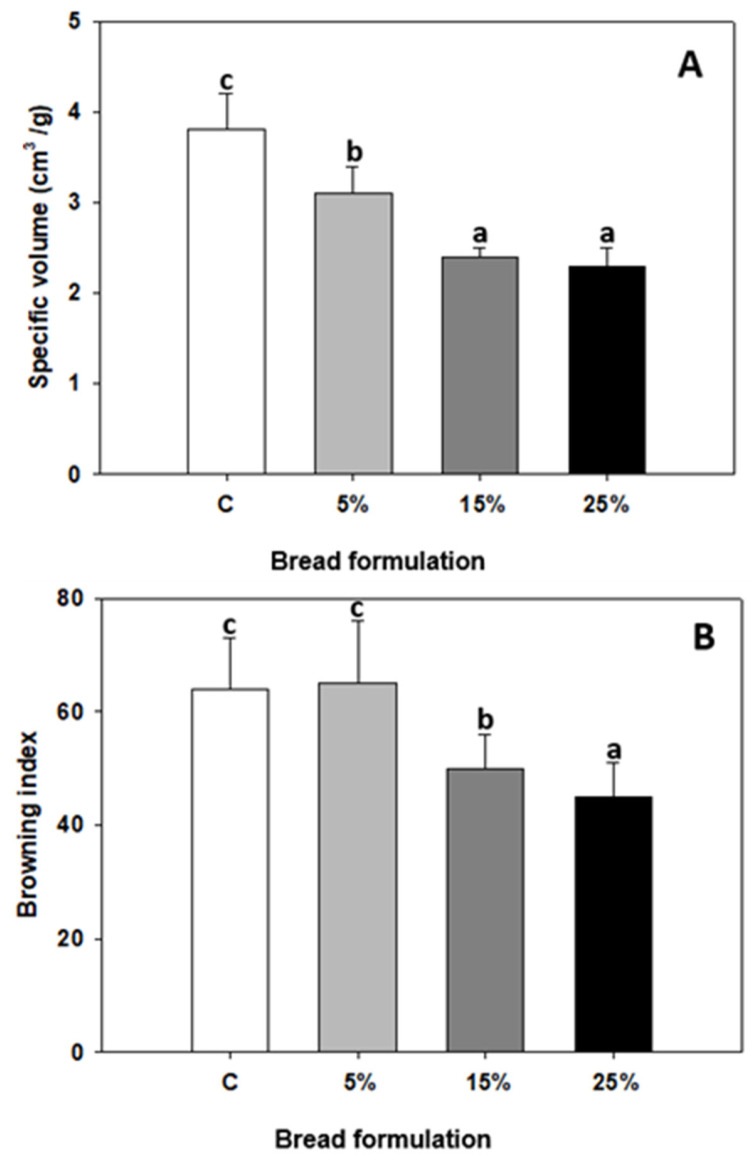
Baking quality parameters: (**A**) specific volume of bread and (**B**) Browning Index. Breads: wheat flour (control: C) and blends of wheat flour and oat flour at 5, 15, and 25%. Different letters in the same figure indicate significant differences (*p* < 0.05).

**Figure 3 foods-12-00170-f003:**
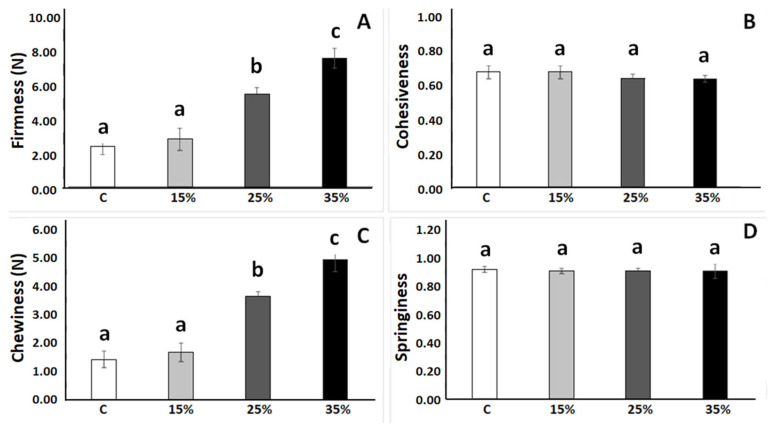
Crumb texture parameters: wheat bread (control: C) and breads with mixtures of oat–wheat flours: 5, 15, and 25%. (**A**). Firmness. (**B**). Cohesiveness. (**C**) Chewiness. (**D**) Springiness. Different letters indicate significant differences between formulations (*p* < 0.05).

**Figure 4 foods-12-00170-f004:**
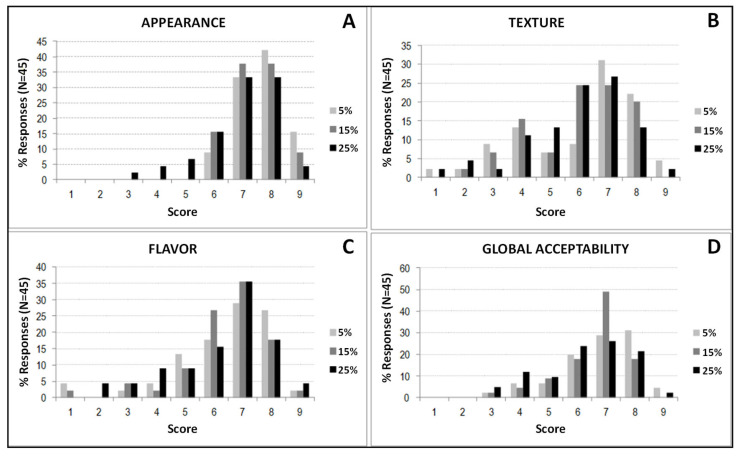
Percentages of responses for breads formulated with 5%, 15%, and 25% of oat flour. (**A**) Appearance. (**B**) Texture. (**C**) Flavor. (**D**) Global acceptability. Attributes: appearance, texture, flavor, overall acceptability.

**Table 1 foods-12-00170-t001:** Formulation of different wheat and oat flour blends.

Ingredients	C	5%	15%	25%
Wheat flour (g)	100	100	100	100
Oat flour (g)	0	5	15	25
NaCl (g)	1.5	1.5	1.5	1.5
Fresh yeast (only bread) (g)	3	3	3	3
Destilled water (mL)	62.4	62.5	64.6	66.7

**Table 2 foods-12-00170-t002:** Alveographic and farinographic parameters of wheat flour (control: C) and blends of wheat flour and oat flour at levels of 5, 15, and 25%.

**ALVEOGRAM**	**Samples**	**P (mmH_2_O)**	**L (mm)**	**P/L**	**W (J × 10^−4^)**
	**C**	136 ± 2 b	67 ± 8 b	2.0 ± 0.3 a	380 ± 35 a
	**5%**	119 ± 2 a	61 ± 11 ab	2.0 ± 0.3 a	310 ± 54 a
	**15%**	158 ± 5 c	59 ± 6 ab	2.7 ± 0.3 b	383 ± 33 a
	**25%**	171 ± 7 d	48 ± 5 a	3.6 ± 0.6 b	344 ± 14 a
**FARINOGRAM**		WA (%)	DT (min)	ST (min)	SD (BU)
	**C**	62.4 ± 0.31 a	31.1 ± 0.9 a	35.9 ± 6.3 a	46 ± 7 c
	**5%**	62.5 ± 0.28 a	29.2 ± 0.6 a	43.1 ± 1.5 a	29 ± 2 bc
	**15%**	64.6 ± 0.35 b	24.8 ± 0.3 a	36.4 ± 3.7 a	16 ± 6 ab
	**25%**	66.7 ± 0.07 c	25.3 ± 4.4 a	28.7 ± 0.6 a	8 ± 5 a

Alveographic parameters: tenacity (P); extensibility (L); tenacity-to-extensibility ratio (P/L), and alveographic work (W). Farinographic parameters: water absorption (WA); development time (DT); stability (ST); and softening degree (SD). Different letters in the same column indicate significant differences between values (*p* < 0.05).

**Table 3 foods-12-00170-t003:** Water activity (aw) and moisture content of dough: wheat flour (control: C) and blends of wheat flour and oat flour at levels of 5, 15, and 25%.

Dough	aw	Moisture (%)
**C**	0.978 ± 0.001 a	55.2 ± 0.1 d
**5%**	0.979 ± 0.002 ab	44.0 ± 0.0 c
**15%**	0.981 ± 0.001 b	42.5 ± 0.2 b
**25%**	0.981 ± 0.001 ab	41.5 ± 0.1 a

Different letters in the same column indicate significant differences between values (*p* < 0.05).

**Table 4 foods-12-00170-t004:** Texture profile analysis of dough: wheat flour (control: C) and blends of wheat flour and oat flour at levels of 5, 15, and 25%.

Texture Parameters						
Doughs	Hard (N)	Cohes (-)	Gumm (N)	Cons (N*s)	Spring (N*s)	Adhes (N*s)
**C**	2.7 ± 0.3 a	0.79 ± 0.02 b	2.1 ± 0.2 a	21.6 ± 1.4 a	0.90 ± 0.01 a	7.8 ± 0.8 b
**5%**	3.8 ± 0.2 b	0.79 ± 0.02 b	3.0 ± 0.2 b	30.1 ± 1.8 b	0.91 ± 0.01 a	9.1 ± 1.3 c
**15%**	5.5 ± 0.5 c	1.05 ± 0.13 c	5.8 ± 0.7 c	41.8 ± 0.9 c	1.53 ± 0.30 b	1.2 ± 0.1 a
**25%**	10.9 ± 0.8 d	0.67 ± 0.03 a	7.3 ± 0.8 d	50.0 ± 1.6 d	2.75 ± 0.05 c	1.2 ± 0.1 a

Texture parameters: hardness (Hard), cohesiveness (Cohes), gumminess (Gumm), consistency (Cons), springiness (Spring), adhesiveness (Adhes). Different letters in the same column indicate significant differences between values (*p* < 0.05).

**Table 5 foods-12-00170-t005:** Parameters from relaxation curves of dough: wheat flour (control: C) and blends of wheat flour and oat flour at levels of 5, 15, and 25%.

Relaxation Parameters					
Dough	E1 (kPa)	E2 (kPa)	E3 (kPa)	T1 (s)	T2 (s)
**C**	0.34 ± 0.03 a	2.79 ± 0.39 a	0.09 ± 0.01 a	275 ± 16 a	7.9 ± 0.6 b
**5%**	0.48 ± 0.04 ab	6.50 ± 0.59 a	0.14 ± 0.06 a	533 ± 1 c	4.2 ± 0.2 a
**15%**	0.60 ± 0.07 b	3.40 ± 0.51 a	0.58 ± 0.10 b	258 ± 23 a	10.0 ± 2.0 b
**25%**	3.42 ± 0.14 c	31.11 ± 7.29 b	2.20 ± 0.17 c	318 ± 55 b	8.4 ± 2.2 b

Elastics moduli (E1, E2, E3) and relaxation times (T1, T2). Different letters in the same column indicate significant differences between values (*p* < 0.05).

**Table 6 foods-12-00170-t006:** Kinetics parameters obtained from modeling curves of dough fermentation. Dough: wheat flour (control: C) and blends of wheat flour and oat flour at 5, 15, and 25%.

	Fermentation Parameters		
Doughs	a (cm^3^)	b (min^−1^)	c
**C**	98.72 ± 12.8 b	0.04 ± 0.01 a	1.81 ± 0.6 a
**5%**	64.90 ± 3.3 a	0.06 ± 0.01 a	2.46 ± 0.6 a
**15%**	60.32 ± 3.5 a	0.06 ± 0.01 a	1.79 ± 0.4 a
**25%**	47.39±12,5 a	0.11 ± 0.03 b	2.10 ± 1.2 a

Different letters in the same column indicate significant differences (*p* < 0.05).

**Table 7 foods-12-00170-t007:** Percentage composition of wheat bread (C) and breads with mixtures of oat–wheat flours: 5, 15, and 25%.

Macrocomponents	BREADS			
	C	5%	15%	25%
**Moisture (%)**	35.49 ± 0.11 c	34.28 ± 0.43 b	32.28 ± 0.33 a	34.92 ± 0.09 c
**Proteins (%)**	9.52 ± 0.08 b	9.36 ± 0.09 b	9.29 ± 0.08 b	8.96 ± 0.05 a
**Lipids (%)**	0.27 ± 0.02 a	0.38 ± 0.02 a	0.70 ± 0.01 b	0.68 ± 0.02 b
**Ash (%)**	1.51 ± 0.02 a	1.35 ± 0.01 a	1.47 ± 0.13 a	1.29 ± 0.01 a
**β-glucans (%)**	0.21 ± 0.04 a	0.32 ± 0.03 b	0.56 ± 0.03 c	0.74 ± 0.00 d
**Total Dietary Fiber (%)**	3.85 ± 0.04 a	5.69 ± 0.10 b	6.00 ± 0.06 c	6.63 ± 0.46 d

Different letters in the same column indicate significant differences between values (*p* < 0.05).

## Data Availability

The data supporting the results of this study are included in the present article.

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
