# Peer review of "High β-Glucans Oats for Healthy Wheat Breads: Physicochemical Properties of Dough and Breads"

_foods, 2022, doi:10.3390/foods12010170_

Round 1

Reviewer 1 Report

This manuscript described the impact of high β-glucans oat on the physicochemical properties of dough and bread. The authors have done a lot of basic work. However, many sentences in the article are too cumbersome to understand and some observations to the manuscript should be resolved.

The detailed comments are as follows:

1. The title is confusing, the “.” may be “:”

2. The Abstract does not need to divide into two paragraphs. Meanwhile, the abstract is too redundant, and the content has to be revised a lot. For example, Lines 14-16 is very confusing.

3. The introduction is too simple, only the nutritional characteristics are introduced preliminarily, but the application of oat or β-glucans in dough or bread and related research reports are not introduced.

4. Lines 103, 106,107, et., “oC” may be a wrong symbol.

5. In the farinographic Assays, the “(TD)” should be corrected as “(DT)”.

6. The Tables should be displayed in a three-line table.

7. Lines 333-339, the references cited may be not connected with this article. The texture of soybean, broad bean, and chickpea-based food in the cited articles was shown in the bread or cooked spaghetti. The texture of dough in this paper and the texture of baked bread cannot be confused.

8. There are too few references in the Results and discussion, so it is difficult to confirm the accuracy of the results. Such as Part 3.2, 3.4, 3.5

9. Part 3.6, it's better to merge the middle paragraphs together.

10. There are too few latest references and much old documents used in this paper, which could not analyze the experimental data and phenomena accurately and reliably.

Reviewer 2 Report

Please see comment and suggestion in the attached file.

Round 2

Reviewer 1 Report

The said changes have been incorporated by the authors; therefore the manuscript can be accepted please.